# Triphenilphosphonium Analogs of Chloramphenicol as Dual-Acting Antimicrobial and Antiproliferating Agents

**DOI:** 10.3390/antibiotics10050489

**Published:** 2021-04-23

**Authors:** Julia A. Pavlova, Zimfira Z. Khairullina, Andrey G. Tereshchenkov, Pavel A. Nazarov, Dmitrii A. Lukianov, Inna A. Volynkina, Dmitry A. Skvortsov, Gennady I. Makarov, Etna Abad, Somay Y. Murayama, Susumu Kajiwara, Alena Paleskava, Andrey L. Konevega, Yuri N. Antonenko, Alex Lyakhovich, Ilya A. Osterman, Alexey A. Bogdanov, Natalia V. Sumbatyan

**Affiliations:** 1Department of Chemistry, Lomonosov Moscow State University, Leninskie Gory 1, 119991 Moscow, Russia; julidev@yandex.ru (J.A.P.); zkh_msu@mail.ru (Z.Z.K.); skvorratd@mail.ru (D.A.S.); bogdanov@belozersky.msu.ru (A.A.B.); 2A.N. Belozersky Institute of Physico-Chemical Biology, Lomonosov Moscow State University, Leninskie Gory 1, 119992 Moscow, Russia; tereshchenkov@list.ru (A.G.T.); nazarovpa@gmail.com (P.A.N.); antonen@belozersky.msu.ru (Y.N.A.); 3Laboratory of Molecular Genetics, Moscow Institute of Physics and Technology, 141700 Dolgoprudny, Russia; 4Center of Life Sciences, Skolkovo Institute of Science and Technology, 143028 Skolkovo, Russia; Dmitrii.Lukianov@skoltech.ru; 5School of Bioengineering and Bioinformatics, Lomonosov Moscow State University, 119992 Moscow, Russia; inna-volynkina@yandex.ru; 6Laboratory of the Multiscale Modeling of Multicomponent Materials, South Ural State University, 454080 Chelyabinsk, Russia; makarovgi@susu.ru; 7Department of Experimental and Health Sciences, Universitat Pompeu Fabra, 08003 Barcelona, Spain; etna.abad@upf.edu; 8Department of Chemotherapy and Mycoses, National Institute of Infectious Diseases, 1-23-1 Toyama, Shinjuku-ku, Tokyo 162-8340, Japan; smurayam@nih.go.jp; 9School of Life Science and Technology, Tokyo Institute of Technology, Yokohama, Kanagawa 226-8501, Japan; kajiwara.s.aa@m.titech.ac.jp; 10Petersburg Nuclear Physics Institute, NRC “Kurchatov Institute”, 188300 Gatchina, Russia; polesskova_ev@pnpi.nrcki.ru (A.P.); konevega_al@pnpi.nrcki.ru (A.L.K.); 11Peter the Great St. Petersburg Polytechnic University, 195251 Saint Petersburg, Russia; 12NRC “Kurchatov Institute”, 123182 Moscow, Russia; 13Institute of Molecular Biology and Biophysics, Federal Research Center of Fundamental and Translational Medicine, 630117 Novosibirsk, Russia; lyakhovich@gmail.com; 14Vall D’Hebron Institut de Recerca, 08035 Barcelona, Spain; 15Genetics and Life Sciences Research Center, Sirius University of Science and Technology, 1 Olympic Ave, 354340 Sochi, Russia

**Keywords:** chloramphenicol, alkyl(triphenyl)phosphonium, bacterial ribosome, molecular dynamics simulations, antibiotic activity, antiproliferative activity

## Abstract

In the current work, in continuation of our recent research, we synthesized and studied new chimeric compounds, including the ribosome-targeting antibiotic chloramphenicol (CHL) and the membrane-penetrating cation triphenylphosphonium (TPP), which are linked by alkyl groups of different lengths. Using various biochemical assays, we showed that these CAM-Cn-TPP compounds bind to the bacterial ribosome, inhibit protein synthesis in vitro and in vivo in a way similar to that of the parent CHL, and significantly reduce membrane potential. Similar to CAM-C4-TPP, the mode of action of CAM-C10-TPP and CAM-C14-TPP in bacterial ribosomes differs from that of CHL. By simulating the dynamics of CAM-Cn-TPP complexes with bacterial ribosomes, we proposed a possible explanation for the specificity of the action of these analogs in the translation process. CAM-C10-TPP and CAM-C14-TPP more strongly inhibit the growth of the Gram-positive bacteria, as compared to CHL, and suppress some CHL-resistant bacterial strains. Thus, we have shown that TPP derivatives of CHL are dual-acting compounds targeting both the ribosomes and cellular membranes of bacteria. The TPP fragment of CAM-Cn-TPP compounds has an inhibitory effect on bacteria. Moreover, since the mitochondria of eukaryotic cells possess qualities similar to those of their prokaryotic ancestors, we demonstrate the possibility of targeting chemoresistant cancer cells with these compounds.

## 1. Introduction

The search for new antimicrobial agents remains a crucial and urgent task, which is largely due to the existence and endless emergence of resistant bacterial strains with various mechanisms of acquired resistance to nearly all clinically relevant antibiotics. These mechanisms include mutations in the drug target site, enzymatic modification or degradation of antibiotics, and active efflux through porins and other permeability barriers [1,2,3].

One promising approach to creating new antibiotics is the development of the so-called twin-drugs–dual-acting compounds, which contain two pharmacophores covalently linked in one molecule. This approach makes it possible to potentially create drugs that are active against drug-resistant microorganisms, have an expanded spectrum of antibacterial activity, compared to the original antibiotics, and have a reduced potential for the generation of bacterial resistance [4]. While each of the two pharmacophores in such a hybrid drug molecule is expected to act independently of the original biological target, the non-cleavable covalent linker tethering the two active moieties endows the drug with a dual mechanism of action. These pharmacophores can be either two antibiotics or an antibiotic with an adjuvant that increases the access of the drug to its intracellular target (e.g., an efflux pump inhibitor or membrane and cell wall-penetrating group, or a moiety that changes the physical properties of the molecule).

Quinolone-based hybrid compounds, especially fluoroquinolones bound to other antibacterial agents, such as oxazolidinones, anilinouracil compounds, tetracyclines, benzylpyrimidine, macrolides, quinolones, oxoquinolysines, or aminoglycosides, are the most studied examples of tethered antibiotics [5,6,7]. Many studies on the development of dual-acting compounds have been conducted on aminoglycosides by linking these molecules to quinolones, as well as to β-lactam antibiotics, CHL, oxazolidinones, or short amphiphilic peptides. Some dual-acting glycopeptide antibiotics have been synthesized, including β-lactam, macrolide moieties, and fragments of natural antimicrobial peptides [6,7,8]. Many of these compounds exhibited a high antibacterial activity not only against Gram-positive strains, but also against Gram-negative bacteria, had a broad spectrum of activity and reduced the toxicity to the mammalian host, compared to the original antibiotics, and were also active against bacterial drug-resistant strains. A number of such hybrid antibiotics have had clinical success in recent years [7].

Another type of dual-acting antibiotics are hybrid compounds containing a component that inhibits efflux pumps, whose mutations are the main cause of intrinsic resistance to the currently available antibiotics against Gram-negative bacteria. This component can be either non-antibacterial [7,9,10] or antibacterial [6,7,11]. For example, aminoquinolones or tobramycin are used to construct dual-acting agents not only as antibiotics, but also as membrane efflux pump inhibitors [11].

Another type of molecule that can be used in the design of dual-acting antibiotics are moieties that provide a better penetration of the antibiotic into bacterial cells. For example, siderophores, which are high-affinity iron chelators produced by bacteria and fungi, have been successfully used in the past to promote active transport of antibiotics into bacterial cells to enhance the action of β-lactam or penicillin antibiotics [7,11].

Moreover, benzoxaboroles are known to enhance the transport of macromolecules into the cell as a result of the interaction with 1,2- and 1,3-diol polysaccharides located on the cell surface. These properties, as well as the ability of some heterocyclic boronic acids and benzoxaborole derivatives to exhibit activity against Gram-negative bacteria with multi-drug resistance, were considered when developing chimeric antibiotics based on glycopeptides or polyene macrocyclic antibiotics containing benzoxaboroles in their structure [6,12] and benzoxaborole derivatives of azithromycin [8].

Triphenylphosphonium (TPP) is a synthetic cation that readily penetrates biological membranes. The positive charge of this moiety is dispersed over the three phenyl residues. As a result, the water dipoles cannot be retained by the cation and, therefore, do not form an aqueous shell preventing the ion from penetrating the membrane using the energy of the transmembrane potential [13]. TPP and its synthetic derivatives have been actively studied mainly as mitochondria-targeting compounds, revealing many interesting properties that can be used to create therapeutic agents [14,15]. In particular, alkyl-TPPs and their derivatives exhibit properties of mild uncouplers of oxidative phosphorylation, with a mechanism comprising the interaction of alkyl-TPP cations with anions of fatty acids, facilitating fatty acid cycling in the membrane [16]. It has also been shown that TPP derivatives exhibit antibacterial properties [17,18,19,20,21,22,23,24,25]. Because of the similarity of bioenergetic processes between bacteria and mitochondria, this effect can be associated both with a decrease in the bacterial membrane potential and with destabilization of the lipid membrane due to a detergent-like effect or the induction of the non-specific membrane permeability at high concentrations of alkyl-TPP derivatives and an increase in the alkyl chain length [17,22,23,26].

It has been reported that along with their antibacterial properties, TPP derivatives of various structures exhibit antiproliferative effects [27,28,29,30]. Most cancer cells possess functional mitochondria, but the oncogenic transformation itself often increases mitochondrial metabolism [31]. The mitochondria of cancer cells can have increased transmembrane potentials in comparison with normal cells [32]. However, the bioenergetics of resistant and cancer stem cells (CSCs), which are mainly responsible for metastasis, differ from the cancer cells themselves [33,34,35]. Ultimately, this may allow cancer cells to be discriminated according to their degree of aggressiveness. The conjugation of TPP with various bioactive compounds [19,29,30] or nanocrystals [36] gives chimeric molecules an increased cellular accessibility and enhances their cytotoxicity against tumor cells. For this reason, TPP conjugates with paclitaxel [37] and doxorubicin [38] have recently been suggested to treat drug-resistant cancers. In turn, several TPP-containing compounds have been used for the elimination of cancer stem cells (CSCs) [39]. A growing body of evidence has now shown that even at low concentrations, some antibiotics can cause mitochondrial dysfunction due to similarities in their structures with bacteria [40,41]. On this basis, some mitochondria-targeting antibiotics have been used as anti-cancer drugs [42,43]. Overall, the synthesis of compounds combining antibiotic derivatives with targeted delivery to mitochondria via a TPP moiety may represent a novel approach in anti-cancer therapy, especially when applied to resistant forms of cancer. 

CHL is a ribosome-targeting antibiotic that binds to the peptidyl transferase center (PTC) [44] of the bacterial ribosome and inhibits peptide bond formation [45]. CHL has been frequently used as a platform to obtain derivatives with an increased potency [46]. This drug is especially amenable to chemical derivatization, because its dichloroacetyl moiety can be easily replaced with a variety of other chemical scaffolds, such as amino acids [47], peptides [48], and an acyl group carrying the polyamine extension [49,50], endowing it with new properties. Thus, the synthesis of novel CHL analogs containing a TPP cation in their structure represents a promising challenge in terms of creating new potential dual-acting antibiotics and antiproliferative agents.

In the current study, we continued our research on the synthesis and exploration of semi-synthetic TPP analogs of CHL [51], i.e., CAM-Cn-TPPs, with the goal of obtaining a new group of CHL derivatives with potentially improved properties. To this end, the dichloromethyl group of the parent CHL compound was replaced with alkyl(triphenyl)phosphonium residues, resulting in CAM-Cn-TPP molecules (Figure 1). Using a competition binding assay, CAM-C10-TPP was shown to exhibit a stronger binding to the bacterial ribosome, compared to CHL, and the new CHL analogs also inhibited protein synthesis in vitro. A toeprinting assay revealed that the mode of action of CAM-Cn-TPP on the bacterial ribosome differs from the site-specific action of CHL, as previously shown for CAM-C4-TPP [51]. While the atomic-resolution structure of the ribosome-bound CAM-C4-TPP compound has been solved and reported recently, possible interactions of the other CAM-Cn-TPP compounds with the *E. coli* ribosome were modeled by molecular dynamics simulations. Using a potential-sensitive fluorescent probe, we found that CAM-C10-TPP and CAM-C14-TPP significantly reduce the membrane potential in *Bacillus subtilis* cells. Experiments with bacteria demonstrated that, in comparison to CHL, CAM-C10-TPP inhibited the growth of the Gram-positive bacteria, *Staphylococcus aureus*, *Listeria monocytogenes*, *Bacillus subtilis*, and *Mycobacterium smegmatis*, to a greater extent. In addition, CAM-C10-TPP and CAM-C14-TPP suppressed some strains of CHL-resistant bacteria. Thus, we showed that CAM-Cn-TPP compounds act both on the ribosome and on bacterial cell membranes, with the TPP fragments of CAM-C10-TPP and CAM-C14-TPP contributing significantly to the inhibitory effect on bacterial growth. We also showed that TPP derivatives are able to target mitochondria in chemoresistant breast cancer cells and derived cancer stem-like cells and reduce their proliferation.

## 2. Results and Discussion

### 2.1. Synthesis of CAM-Cn-TPPs

We have recently synthesized and extensively characterized the first semi-synthetic TPP analogs of CHL, CAM-C4-TPP [51]. In the current study, we continued our research on the TPP analogs of CHL, synthesized two more of them, CAM-C10-TPP and CAM-C14-TPP, and explored their inhibitory properties for the potential development of novel antimicrobials, as well as antiproliferative agents.

The dual-action mechanism of these compounds should be mediated, on the one hand, by their binding and action on the bacterial ribosome (similar to the action of CHL), and, on the other hand, by their action on bacterial membranes (similar to alkyl-TPP salts). For this purpose, the dichloromethyl group of CHL was replaced with an alkyl-TPP moiety, resulting in CAM-Cn-TPP molecules (Figure 1).

These compounds were designed with the idea that the amphenicol moiety would anchor the compound in the canonical CHL binding site within the PTC of the bacterial ribosome, and an additional group would form multiple interactions with the walls of the nascent peptide exit tunnel (NPET). We chose TPP as such a group because its positive charge, delocalized over the relatively large hydrophobic surface of the benzene rings, can provide non-specific interactions with negatively charged phosphates of the 23S rRNA, and its three phenyl rings may be available for stacking with nucleobases. In the case of CAM-C4-TPP [51], we showed how this analog binds to the bacterial ribosome and inhibits bacterial protein synthesis, both in vitro and in vivo. Moreover, according to our rational design hypothesis, the TPP group in these compounds should allow CAM-Cn-TPPs into bacterial cells, since the TPP itself is known to be a membrane-penetrating cation [13].

As for the interactions with the ribosome, we expected that linkers of variable lengths connecting the two terminal parts of these compounds (CAM and TPP) would ensure the optimal binding of CAM-Cn-TPPs to the ribosome and enable similar non-specific interactions with rRNA nucleotides at different depths of the NPET, as observed in the X-ray crystal structure of the *Thermus thermophilus* 70S ribosome in the complex with the CAM-C4-TPP [51]. The choice of linker length was also based on the data on the inhibition of bacterial growth by alkyl-TPPs, where it was shown that the toxic effect on different bacterial species increases with increasing lipophilicity, and this effect is related to the different permeability of bacterial coats for alkyl-TPPs [17]. In addition, we modeled the length of the linkers using in silico simulations. Linkers of lengths 10 (C10) and 14 (C14) from the methylene groups were chosen. In the latter case, an amide group was introduced to reduce any possible side effects associated with the high lipophilicity of the resulting compound.

The synthesis of CAM-C10-TPP and CAM-C14-TPP was performed by the acylation of chloramphenicol amine (CAM) with carboxyl derivatives of TPP using succinimide ester (Figure 1), which is similar to the synthesis of CAM-C4-TPP [51]. The chemical structures of the obtained CAM-Cn-TPP molecules were confirmed by mass-spectrometric analysis, as well as by 1H-, 13C-, and 31P-NMR.

### 2.2. CAM-Cn-TPPs Bind to the Bacterial Ribosome with Different Affinities and Inhibit Protein Synthesis, Allowing the Formation of Short Peptides

All new semi-synthetic CAM-Cn-TPP compounds were expected to bind and act on bacterial ribosomes by inhibiting protein synthesis, which is similar to CAM-C4-TPP and the PTC-targeting parent antibiotic CHL. To assess the affinity of CAM-C10-TPP and CAM-C14-TPP for the bacterial 70S ribosome (Figure 2A), a competition-binding assay exploiting BODIPY-labeled erythromycin (BODIPY-ERY) was used [47,52,53]. CAM-C10-TPP was found to have significantly greater (~7-fold) affinity to the ribosome, compared to the parent CHL, and a slightly greater affinity (~1.5-fold), compared to CAM-C4-TPP (K_Dapp_ = 0.4 ± 0.04 µM for CAM-C10-TPP vs. 2.8 ± 0.4 µM for CHL; and K_Dapp_ = 0.61 ± 0.07 µM for CAM-C4-TPP [51]). Unexpectedly, CAM-C14-TPP binds to the 70S ribosome with a significantly lower affinity, compared to other CAM-Cn-TPPs (~60-90-fold) and CHL (~13-fold, K_Dapp_ = 36 ± 9 µM for CAM-C14-TPP). It is likely that in the case of CAM-C14-TPP, the linker may be too long, preventing the optimal positioning of the TPP moiety in the NPET. Thus, CAM-C10-TPP and CAM-C14-TPP bind to the bacterial ribosome. However, their affinity strongly depends on the length of the linker connecting the two chromophores. Among the three CAM-Cn-TPPs, the C10-linker was optimal in terms of its ability to bind to the 70S ribosome.

Next, we tested the effects of CAM-C10-TPP and CAM-C14-TPP in bacterial cells using an *E. coli* ∆*tolC*-based reporter system, which is designed to screen inhibitors targeting either protein synthesis or DNA replication [54]. In this reporter system, the gene of the far-red fluorescent protein, *katushka2S*, is inserted downstream of the genetically modified tryptophan attenuator, making it possible to express Katushka2S only upon exposure to ribosome-stalling compounds (Figure 2B, CHL and ERY, red pseudocolor rings). The red fluorescent protein gene *rfp* is placed under the control of the SOS-inducible sulA promoter, allowingthe expression of the Red Fluorescent Protein (RFP) reporter to be determined under the action of appropriate compounds such as DNA gyrase inhibitors (e.g., levofloxacin, LEV, Figure 2B, green pseudocolored rings).

We used chloramphenicol amine (CAM), N-acetyl-chloramphenicol amine (CAM-Ac), and alkyl-TPP (C10-TPP and C14-TPP) as negative controls in this assay. We observed neither the induction of either of the two reporters (no colored rings) nor the inhibition of cell growth by CAM or CAM-Ac (no dark zone in the middle). This indicates that the removal of the two Cl atoms from the dichloroacetyl group of CHL makes it inactive [55]. The C10-TPP and C14-TPP compounds (lacking the CAM moiety) inhibit bacterial cell growth (dark area in the middle) but do not induce either of the two reporters, suggesting that they act through an entirely different cellular target (likely targeting the membrane [17]). For CAM-C10-TPP and CAM-C14-TPP, as well as for the positive controls, CHL and ERY, red pseudocolor rings are observed due to the expression of the fluorescent protein, Katushka2S (but not RFP), indicating that these compounds specifically inhibit protein synthesis, as was previously observed for CAM-C4-TPP [51].

The next step was to test the new CAM-Cn-TPP compounds for their ability to inhibit the synthesis of the firefly luciferase reporter in vitro in a cell-free transcription-translation system based on an *E. coli* S30 extract. CAM-C10-TPP, like CAM-C14-TPP, inhibits bacterial translation, which is similar to CHL (Figure 2C) and CAM-C4-TPP [51]. At the same time, both CAM-C10-TPP and CAM-C14-TPP had no effect on eukaryotic in vitro translation (Figure 2C), which was revealed by a similar approach using the eukaryotic in vitro translation system.

Recently, it was shown that the mechanism of action of CHL on translation is different from what was previously widely accepted (blocking the accommodation of the incoming aminoacyl-tRNA in the PTC): in the presence of CHL, short peptides can be synthesized on the ribosome, and the action of the antibiotic is context-specific [56]. CHL arrests translation only when alanine and, to a lesser extent, serine or threonine appear in the penultimate position (E site) of the growing polypeptide chain and only if there is no glycyl-tRNA in the A site of the ribosome [56]. As shown in our recent study, the action of CAM-C4-TPP in bacterial translation was also context-specific, but its mode of action was different from the site-specific action of CHL [51].

To investigate the mechanism of action of CAM-C10-TPP and CAM-C14-TPP in the translation process, a primer extension inhibition assay (toeprinting) was used. This method allows for the unambiguous identification of drug-induced ribosome stalling sites along the mRNA, with single-nucleotide precision [57]. This technique is also used to determine the context-specificity of antibiotic action [56]. For this experiment, *trpL* mRNAs were chosen as the template, as in the pDualrep2 reporter system. In contrast to CHL, which stalls ribosomes at the Ile4 codon (Figure 2D and Appendix A, blue arrowhead) of the corresponding Ala3 in the penultimate position of the growing polypeptide chains (E-site), CAM-C10-TPP, CAM-C14-TPP, and CAM-C4-TPP blocked the ribosome progression at Val6 (Figure 2D, red arrowhead) and slightly blocked it at Ile4. Compared to the results obtained previously for CAM-C4-TPP using *rst1* and *rst2* mRNA templates [51], more unambiguous results were obtained using a *trpL* mRNA template. The main drug-induced ribosome stalling sites are different for CHL and CAM-Cn-TPP, and there are also CAM-Cn-TPP-specific stalling positions. These results support the conclusion that CAM-Cn-TPPs have an idiosyncratic mode of action and a unique context-specificity that differs from that of the original CHL [51].

### 2.3. Possible CAM-Cn-TPPs Interaction Dynamics during Translation

To assess the possible interactions of CAM-Cn-TPPs with the ribosome during translation, we modeled the structures of complexes of the new compounds, CAM-C10-TPP and CAM-C14-TPP, with the *E. coli* 70S ribosome using molecular dynamics simulations. To this end, we used the structure of the *E. coli* 70S ribosome in the classical non-rotated a/A-p/P-state, which is conformationally similar to the ribosome containing tRNAs during translation [58]. Analysis of the most populated clusters (Figure 3, Appendix A) for CAM-Cn-TPP complexes revealed that all these compounds are able to interact quite stably with the non-canonical CHL binding site [59]. The CAM-C10-TPP complex is characterized by the most stable stacking interactions and hydrogen bonding and is similar to the analogous CHL complex. Obviously, the positive charge of the TPP fragment is responsible for the nonspecific affinity of all CAM-Cn-TPPs to the negatively charged rRNA. 

CAM-Cn-TPPs were constructed as bidentate ligands, differing in the length of the flexible alkyl chain linking CAM- and TPP- moieties. The synergism of the interactions of these two basic structural elements with one or the other region of the NPET depends on the length of the CAM-Cn-TPP linker. Thus, the relatively long and flexible alkyl chain of CAM-C10-TPP allows the CAM and TPP residues to adapt to their binding sites in the NPET (Figure 3A,B). On the contrary, the longer linker in the CAM-C14-TPP structure results in the inability of this compound to fit snugly in the space between the optimally bound CAM and TPP fragments. These findings may explain the higher affinity of the CAM-C10-TPP to the bacterial ribosomes, compared to the CAM-C14-TPP.

It is noteworthy that models of CAM-Cn-TPPs complexes with the *E. coli* ribosome in the canonical A,A/P,P–state described above are in agreement with the foregoing data on translation arrest in the presence of CAM-C10-TPP and CAM-C14-TPP (Figure 2D) or CAM-C4-TPP, as described in [51]. For all three mRNA templates, *trpL* for CAM-C10-TPP and CAM-C14-TPP, and *trpL*, *rst1*, and *rst2* for CAM-C4-TPP, the peptides synthesized before arrest contain sequences corresponding to the −2 to −5 codon regions (2 to 5 amino acid residues from the C-terminus of the nascent peptide), consisting of amino acids mainly with hydrophobic and aromatic side groups, which may be near CAM-Cn-TPP during translation in NPET. These positions are occupied by the A−4IF−2 in the case of *trpL*, W−5VT−3 in the case of *rst1*, and F−5AI−3 in the case of *rst2* templates. Obviously, these amino acids are capable of forming hydrophobic contacts with the TPP residue of CAM-Cn-TPP, so that the nascent peptide can bind firmly in the NPET, thus hindering translation.

### 2.4. CAM-Cn-TPP Cause a Decrease in the Membrane Potential of B. subtilis

As noted in the Introduction, alkyl-TPPs and their derivatives exhibit an antibacterial effect, which is associated with a decrease in the membrane potential of bacteria [17,22,23]. Thus, we further examined the effect of CAM-Cn-TPP on the bacterial membrane potential of *B. subtilis*. In particular, the membrane potential of *B. subtilis* can be estimated from the fluorescence of the potential-sensitive dye, DiS-C3-(5), by measuring the changing fluorescence. The potential-dependent accumulation of the dye inside the bacterial cell causes quenching and leads to a decrease in fluorescence. Under the influence of substances that reduce the membrane potential, the dye is released and the fluorescence increases. As shown in Figure 4, submicromolar concentrations of CAM-C10-TPP and CAM-C14-TPP caused a decrease in the membrane potential of *B. subtilis* on a minute time scale, whereas 10 µM of CAM-C10-TPP or CAM-C14-TPP caused a rapid drop in the membrane potential to the level observed with the channel-forming antibiotic gramicidin A, which is known to cause the membrane potential of bacteria to vanish, whereas CAM-C4-TPP only takes effect at high (10 µM or more) concentrations. Therefore, the action of CAM-Cn-TPP may have a double effect on bacterial cells and be based on the depolarization of the bacterial membrane by analogy with the mechanism of action of Cn-TPP [17].

### 2.5. CAM-C10-TPP and CAM-C14-TPP Inhibit Bacterial Growth

In order to check whether the synthesized compounds are antimicrobial agents, we tested their action on a number of bacterial species (Table 1, Table 2 and Appendix A). Both CHL and alkyl-TPPs are known to act on Gram-positive bacteria [17,49]. CAM-C10-TPP and CAM-C14-TPP were found to be able to inhibit the growth of the *S. aureus*, *L. monocytogenes*, *B. subtilis*, and *Mycobacterium sp.* strains. Moreover, CAM-C10-TPP suppressed the bacterial growth more effectively than CHL, depending on the strain (Table 1). CAM-C10-TPP was also found to inhibit CHL-resistant *E. coli* strains (Appendix A).

These findings prompted us to further investigate the antibacterial activity of CAM-Cn-TPP on CHL-resistant strains of *B. subtilis* and *E. coli*, with the *tolC* genes deleted, which are more sensitive to the action of compounds that might otherwise be pumped out of the Gram-negative bacterial cells by the TolC efflux pump. AcrAB-TolC is the main multi-drug resistance transporter of Gram-negative bacteria, which is responsible for the efflux of C10-TPP derivatives [18,60]. Apparently, the efflux of CAM-Cn-TPP is also mediated by this transporter, but the degree of involvement of other TolC-containing transporters in CAM-Cn-TPP efflux has not yet been clearly defined.

*E. coli* ∆*tolC* CHL-resistant strain harboring plasmid encoding for chloramphenicol acetyltransferase (*cat*) *(pCA24N-LacZ)* and the control *E. coli* ∆*tolC* strain was used to evaluate the inhibitory effect of the compounds. CHL-resistant strains of *B. subtilis pHT01-cat* and *B. subtilis pHT01-cfr* were prepared by means of a transformation of a *B. subtilis 168* strain with plasmids harboring the *cat* and *cfr* genes (chloramphenicol-florfenicol resistance) gene. The *cfr* gene encodes for the methyltransferase, which catalyzes the methylation of m^2^A2503 in the 23S rRNA and causes resistance to a variety of ribosome-targeting antibiotics that bind in the A site of the bacterial ribosome [61].

As follows from the data in Table 2, CAM-C10-TPP inhibited the growth of *E. coli* ∆*tolC* strain with the same efficiency as CHL, and CAM-C14-TPP inhibited it with a slightly lower efficiency. In contrast, CAM-Cn-TPPs were significantly more effective against CHL-resistant strains in comparison with CHL. At the same time, the MIC values for Cn-TPP on all tested strains were lower than those for CAM-Cn-TPPs. A similar effect was observed for CHL-resistant *B. subtilis* because of the presence of the *cat* or *cfr* genes. While CAM-Cn-TPPs clearly inhibit protein biosynthesis in bacteria in vitro, damage to bacterial membranes due to the presence of an alkyl-TPP fragment in their structure contributes more to the action of CAM-Cn-TPPs at the cellular level.

If we compare the TPP analogs of CHL with the nearest structural analogs exhibiting antimicrobial activity, in which two pharmacophores are conjugated through the alkyl linker, we can note that the antibacterial effect of these compounds is due to the fact that they either bind to bacterial ribosomes or inhibit translation in bacteria, like polyamide analogs of CHL [49,50], or disrupt the membrane potential of bacteria, since the alkyl TPP conjugates with fluorescein [22,23] or coumarin [20,21], while CAM-Cn-TPPs can exhibit both these effects.

### 2.6. CAM-C10-TPP and CAM-C14-TPP Show a Reduced Toxicity Compared to Alkyl-TPP on Mammalian Cells

As we have already shown, CAM-Cn-TPPs have no noticeable effect on the eukaryotic translation process (Figure 2C). Given the obvious antibacterial activity of CAM-C10-TPP and CAM-C14-TPP, it was important to assess their cytotoxicity for mammalian cells. The Mosman (“MTT”) assay was used for this purpose [62]. 

CAM-C10-TPP and CAM-C14-TPP are more toxic to various eukaryotic cell lines in comparison to CHL (Table 3), but at the same time, they are significantly less toxic than the well-known cytotoxic drug doxorubicin, which we used as a highly toxic control. Notably, CAM-Cn-TPPs were less toxic than the corresponding alkyl-TPPs, which are part of the molecular structures of these analogs, and were also used (as bromides) as control compounds. Alkyl-TPPs are known to be quite toxic, and the toxicity effect seems to be mainly due to their ability to accumulate in mitochondria [63,64]. On the other hand, this effect may be caused by a decrease of cellular metabolism under the influence of TPP derivatives, which may be a consequence of a charge change on the cell membrane, i.e., the same effect that provides them with antibacterial properties [24,25,26]. 

The relatively high toxicity of CAM-Cn-TPP to human adenocarcinoma cells (MCF7 and A549, Table 3) may indicate that these compounds can be used as antiproliferative agents. CAM-C10-TPP and CAM-C14-TPP have approximately 4–2 times the selectivity of action for A549 and noncancerous VA13 cell lines. These results are consistent with previous reports [65,66] that showed that delocalized lipophilic cation-containing compounds may have selective cytotoxicity against cancer cells. Otherwise, their selectivity against cancer cell lines was lacking, compared to the HEK293T cells of a noncancerous etiology, but they are compatible with the growth rate of cancer cells.

### 2.7. Anticancer Activities of CAM-C10-TPP and CAM-C14-TPP

We recently provided in vivo [67] and in vitro [68] evidence that certain bactericidal antibiotics suppress tumor growth by inducing mitochondrial dysfunction. Our results showed that antibiotics preferentially suppress the growth of cancer chemoresistant and the so-called cancer stem cells (CSC), two major categories of cells responsible for cancer recurrence and metastasis [41]. These cells differ from the corresponding parental cells and display a different response to the same microenvironmental stimuli, allowing them to reduce their proliferating rate and survive chemotherapeutic treatment. Here, we tested the abovementioned CAM-TPP derivatives on chemoresistant triple negative breast cancer (TNBC) cell models, as well as on corresponding CSC-like spheroid cells. We created several relevant models based on TNBC cell lines, particularly MDA-MB-231 and BT-549, to determine their resistance to cyclophosphamide. To create CSC-like cells, parental cells were resuspended in nonadherent conditions to form tumorspheres. We applied TPP derivatives and found that both CAM-C10-TPP and CAM-C14-TPP, but not CAM-C4-TPP, significantly reduced the proliferation of TNBC cell lines (Figure 5). We also noticed that higher concentrations of derivatives preferentially inhibited the proliferation of chemoresistant rather than parental cancer cells. 

Several TPP-containing compounds have been proposed for the elimination of CSCs by the Lisanti group [39]. In the current work, we studied the effect of CAM-TPP derivatives on CSC-like TNBC cell models. We found a significant decrease in spheroid formation when exposed to CAM-C10-TPP (also to CAM-C14-TPP) (Figure 6A,E). The same compounds reduced the survival of CSCs formed from parental and resistant TNBC cell lines (Figure 6B–D,F–H).

We also compared the effects of the obtained compounds on the suppression of normal cell growth. The preliminary results with CAM-C10-TPP showed that normal human fibroblasts were less affected by submicromolar concentrations of CAM-TPP derivatives than cancer cells (Appendix A). This suggests a pharmacological window to discriminate between healthy and cancer cells.

It should be noted that antiproliferative activity was also detected in TPP and Cn-TPP itself, which seems to be related to the depolarization of the mitochondrial membrane potential [27,28,29,30]. In this regard, we consider these data on the use of CAM-Cn-TPP derivatives as one of the possible approaches to anti-cancer therapy, requiring more careful study and titration of concentrations and doses. Overall, our data suggest that specific targeting of cancer cell mitochondria with dual-acting compounds may have clinical advantages for drug development against resistant forms of cancer.

## 3. Materials and Methods

### 3.1. Chemicals and Materials

The following reagents were used: chloramphenicol (Sigma, Steinheim, Germany), 1-hydroxysuccinimide (HOSu), *N,N*′-dicyclohexylcarbodiimide (DCC), 4-aminobutyric acid (GABA), triphenylphosphine, and 11-bromundecanoic acid (Sigma-Aldrich Chemie GmbH, Steinheim, Germany). The fluorescent erythromycin derivative, BODIPY-Ery, was obtained previously [69]. The alkyl-TPPs were obtained according to [17].

### 3.2. Synthetic Procedures

The scheme for the synthesis of chloramphenicol triphenyphosphonium analogs (CAM-C10-TPP and CAM-C14-TPP) is represented in Figure 1. (1R,2R)-2-amino-1-(4-nitrophenyl)propane-1,3-diol (chloramphenicol amine, CAM, 2) was obtained via the acid hydrolysis of chloramphenicol (CHL, 1), according to a procedure [55] described in [47,53]. (10-carboxydecyl)(triphenyl)phosphonium bromide (5) was obtained by the condensation of 11-bromoundecanoic acid (**3**) and triphenylphosphin (4) for 12 h at 85 °C.

*(N-{[(1R,2R)-1,3-dihydroxy-1-(4-nitrophenyl)propan-2-yl]amino}-11-oxoundecyl)(triphenyl)phosphonium bromide (CAM-C10-TPP)*. To a cold solution of 140 mg (0.25 mmol) of (10-carboxydecyl)(triphenyl)phosphonium bromide (5) and 30 mg (0.25 mmol) of HOSu in 5 mL of anhydrous CH_2_Cl_2_, 62 mg (0.3 mmol) of DCC was added at 0 °C. The mixture was stirred for 2 h at 0 °C and overnight at RT. The formed precipitate was filtered off, and the solvent was removed in vacuo. The residue was dissolved in 1 mL of DMF, then 62.5 mg (0.25 mmol) of CAM (2), and 52.8 µL (0.375 mmol) of DIPEA in 250 µL of DMF was added, and the resulted mixture was stirred for 5 h at RT and overnight at 4 °C. Then, the reaction mixture was diluted with 15 mL of water, and 1N aqueous HCl was added dropwise to a neutral pH. The mixture was then extracted with CHCl_3_ (3 × 15 mL), and the combined organic extracts were washed with water (3 × 10 mL). The organic layer was dried over anhydrous Na_2_SO_4_, and the volatiles were evaporated in vacuo. The target product was isolated from residue by purification on a silica gel column eluting with a solvent system of CHCl_3_:MeOH, 6:1. Yield: 123 mg (73%); TLC: *R*_f_ (CHCl_3_:MeOH, 6:1) 0.3, *R*_f_ (CH_3_Cl:MeOH, 9:1) 0.26; LC-MS *m/z* calculated for C_38_H_46_N_2_O_5_P (M)^+^: 641.3, found 641.4; *t*_R_ = 1.08 min; and ESI-MS *m/z* calculated for C_38_H_46_N_2_O_5_P (M)^+^: 641.3133, found 641.3149.

*N-[(1R,2R)-1,3-dihydroxy-1-(4-nitrophenyl)propan-2-yl]-4-(triphenyl)phosphoniumundecanamidobutamide bromide (CAM-C14-TPP)* was obtained as CAM-C10-TPP from 263 mg (0.5 mmol) of (10-carboxydecyl)(triphenyl)phosphonium bromide (**5**), 58 mg (0.5 mmol) of HOSu, 103 mg (0.5 mmol) of DCC, 200 mg (0.5 mmol) of GABA-CAM·TFA (**6**), and 253 µL (1.42 mmol) of DIPEA. The target product was isolated on a silica gel column eluting with a solvent system of CHCl_3_:MeOH:NH_4_OH = 65:25:4. Yield: 261 mg (65%); TLC: *R*_f_ (CHCl_3_:MeOH, 5:1) 0.53; *R*_f_ (CHCl_3_:MeOH:NH_4_OH, 65:25:4) 0.7; LC-MS *m/z* calculated for C_42_H_53_N_3_O_6_P (M)^+^: 726.4, found 726.7; *t*_R_ = 0.50 min; and ESI-MS *m/z* calculated for C_42_H_53_N_3_O_6_P (M)^+^: 726.3667, found 726.3692.

See also the Appendix A for more detailed procedures, characteristics, and NMR-data.

### 3.3. In Vitro Binding Assay

The binding affinity of CAM-Cn-TPP to *E. coli* ribosomes was analyzed by a competition-binding assay using the fluorescently-labeled BODIPY-ERY, as described before [47,52,53,69]. BODIPY-Ery (16 nM) was incubated with the ribosomes (50 nM) in the buffer containing 20 mM HEPES-KOH (pH 7.5), 50 mM NH_4_Cl, 10 mM Mg(CH_3_COO)_2_, 4 mM mercaptoethanol, and 0.05% Tween-20 for 30 min at 25 °C. Solutions of CHL, CAM-Cn-TPP, and CnTPP in different concentration ranges were added to the formed complex. The mixture was incubated for 2 h, until an equilibrium was reached, and the values of fluorescence anisotropy were measured by VICTOR X5 Multilabel Plate Reader (Perkin Elmer, Waltham, MA, USA). The dissociation constants were calculated based on the assumption of the equilibrium competitive binding of two ligands at a single binding site, as described in [70].

### 3.4. Detection of the Translation Inhibitors with a pDualrep2 Reporter

Reporter strain *JW55035* [71] Δ*tolC (BW25113) pDualrep2* was used, as previously described [72]. The tested antibiotics, CAM-C10-TPP (10 mM, 1.5 μL), CAM-C14-TPP (10 mM, 1.5 μL), C10-TPP (10 mM, 1.5 μL), C14-TPP (10 mM, 1.5 μL), CAM (10 mM, 1.5 μL), CAM-Ac (10 mM, 1.5 μL), chloramphenicol (CHL, 2 mM, 1 μL), erythromycin (Ery, 7 mM, 1 μL), and levofloxacin (Lev, 70 nM, 1 μL), were applied to an agar plate that already contained a lawn of the reporter strain. After being incubated overnight at 37 °C, the plate was scanned by ChemiDoc (Bio-Rad) in the modes, “Cy3-blot” for RFP and “Cy5-blot” for Katushka2S.

### 3.5. In Vitro Translation Inhibition Assay and Toeprinting Assays

The inhibition of firefly luciferase synthesis in cell-free translation systems by CAM-Cn-TPP was tested with an *E. coli* S30 Extract System for Linear Templates (Promega, Madison, MI, USA). The reactions were programmed with 100 ng of mRNA and were carried out in 5 μL aliquots at 37 °C for 30 min. The activity of in vitro synthesized luciferase was assessed using 5 μL of the substrate from the Steady-Glo Luciferase Assay System (Promega).

The inhibition of eukaryotic translation was measured in Rabbit Reticulocyte Lysate (Promega) according to the manufacturer’s protocol. The reactions were programmed with 100 ng of Fluc mRNA and were carried out in 5 μL aliquots at 37 °C for 30 min. The activity of the in vitro synthesized luciferase was assessed using 5 μL of the substrate from the Steady-Glo Luciferase Assay System (Promega).

The toeprinting assay was carried out using *trpL* mRNA as a template, as described in [73], and the reactions were preincubated for 5 min with 30 μM of the tested compound.

### 3.6. Bacteria Inhibition Assays

#### 3.6.1. Bacterial Strains

To prepare the bacterial suspension, bacterial stock cultures were sub-cultured onto plates with the proper agar medium and incubated overnight at 30 °C or 37 °C, until reaching the optical density of 1.5 (at 600 nm), which was measured on a Varioskan LUX microplate reader (Thermo Scientific, Waltham, MA, USA) or an Ultrospec 1100 pro spectrophotometer (Amersham Biosciences Corp., Piscataway, NJ, USA).

Standard laboratory strains of *Bacillus subtilis* subs. *subtilis* Cohn 1872, stains *BR151*, and *168*, *Staphylococcus aureus* subsp. *aureus* Rosenbach 1884 strains *JCM 2151* and entry MC#144 (from the Microorganisms Collection of the Moscow State University), *Listeria monocytogenes* Pirie 1940, *Mycobacterium smegmatis* Lehmann and Neumann 1899 (MC#377), and *Escherichia coli* Castellani and Chalmers 1919, strain *JW5503* (with the deletion of *tolC* gene), which are resistant to the chloramphenicol strains, *J53rif*, *C600rif/pIB55-1*, *C600rif/pIP162-1*, and *C600recA naI*, were used.

*S. aureus* was grown in Bacto Tryptic Soy Broth, *L. monocytogenes* in Brain Heart Infusion Broth, and *E. coli* in LB. Bacterial cells were grown at 30 °C or 37 °C in the appropriate medium at a 140 rpm shaking frequency.

A standard laboratory strain of *E. coli JW5503*, with the deletion of the *tolC* gene, referred to here as the ∆*tolC* strain, was used to obtain the ∆*tolC pCA24N-LacZ* strain by means of transformation with the plasmid pCA24N-LacZ-harboring chloramphenicol acetyltransferase (*cat*) gene.

A standard laboratory strain of *B. subtilis 168* was used to obtain the CHL-resistant strains, *B. subtilis pHT01-CAT* and *B. subtilis pHT01-cfr*, by means of transformation with the corresponding plasmids. Competent cell preparation and transformation procedures were conducted, as reported in [74,75]. The desired colonies were selected at 10 µg/mL of CHL for *B. subtilis pHT01* and 5 µg/mL or 3.2 µg/mL of CHL for *B. subtilis pHT01-cfr*.

#### 3.6.2. Plasmids

The pcan24N-lacZ was purified from the *JW0335* strain (ASKA-collection), *pHT01* (was kindly provided by Dr. Svetlana Dubiley), and the pHT01-cfr plasmid was obtained by the following procedure. The whole pHT01 plasmid sequence was amplified with the primers, 5′-TTGATATGCCTCCTAAATTTTTATC-3′ and 5′-TATGAGATAATGCCGACTG-3′. The *cfr* gene was amplified with the primers, 5′-acagtcggcattatctcataCTATTGGCTATTTTGATAATTACC-3′ and 5′-aaatttaggaggcatatcaaATGAATTTTAATAATAAAACAAAGTATGG-3′, using the *Staphylococcus sp. (cfr+)* genome DNA as a template. The joining of the two DNA fragments was performed with the NEBuilder HiFi DNA Assembly Master Mix (NEB), and subsequently, the right clones were selected. 

#### 3.6.3. CAM-Cn-TPP-Dependent Bacterial Growth Suppression Screening of TolC-Requiring Transporters

The *E. coli* deletion mutants’ panel [18,60] was selected. The selected bacterial strains belonging to the panel were diluted in fresh LB media after overnight growing, and 200 µL of bacterial cell cultures (5 × 10^5^ cells/mL) were inoculated into 96-well plates. The preselected CAM-C14-TPP and CAM-C10-TPP concentrations (5 µM, 10 µM, and 20 µM) were added to each mutant, and the optical density at 620 nm was measured using a Thermo Scientific Multiskan FC plate reader. The cells were left to grow for 21 h at 37 °C, and the optical density at 620 nm was measured. All experiments were performed at least in triplicates.

#### 3.6.4. MIC Determination

The MICs for CAM-Cn-TPP and Cn-TPP were determined by Mueller-Hinton broth microdilution, as recommended by CLSI in the Methods for Dilution Antimicrobial Susceptibility Tests for Bacteria that Grow Aerobically, Approved Standard, 9th ed., CLSI document M07-A9, using in-house-prepared panels. The compounds were diluted in a 96-well microtiter plate to final concentrations ranging from 0.5 to 360 μM in a 250-µL aliquot of the bacterial suspension, followed by incubation at 37 °C or 30 °C for 18 h. The MIC was determined as the lowest concentration that completely inhibited bacterial growth. The bacterial growth was observed visually alongside OD measurements. The experiments were carried out in triplicate. 

### 3.7. Measurement of the B. subtilis Membrane Potential

The membrane potential of *B. subtilis* was estimated by measuring the fluorescence of the potential-dependent probe, DiS-C3-(5) [76]. *B. subtilis* from the overnight culture were seeded into a fresh LB medium, followed by growth for 24 h, until reaching the optical density of 0.8 at 600 nm. Then, the bacteria were diluted 20-fold in a buffer containing 100 mM of KCl and 10 mM of Tris, pH 7.4. The fluorescence was measured at 670 nm (excitation at 630 nm) using a Fluorat-02-Panorama fluorimeter.

### 3.8. In Vitro Survival Assay (MTT Assay)

The cytotoxicity of the tested substances was tested using the MTT (3-(4,5-dimethylthiazol-2-yl)-2,5-diphenyltetrazolium bromide) assay [62], with some modifications, and 2500 cells per well for the MCF7, HEK293T, and A549 cell lines or 4000 cells per well for the VA-13 cell line were plated in 135 µL of DMEM-F12 media (Gibco, Waltham, MA, USA) in a 96-well plate and incubated in the 5% CO_2_ incubator for first 16 h, without treatment. Then, 15 µL of media-DMSO solutions of the tested substances was added to the cells (the final DMSO concentrations in the media were 1% or less) and treated cells for 72 h with 25 nM–50 µM (eight dilutions) of our substances (triplicate each), and serial DMSO dilutions and doxorubicin were used as controls. The MTT reagent (Paneco LLC, Moscow, Russia) was then added to cells to a final concentration of 0.5 g/L (10× stock solution in PBS was used) and incubated for 2.5 h at 37 °C in the incubator under an atmosphere of 5% CO_2_. The MTT solution was then discarded, and 140 µL of DMSO (PharmaMed LLC, Krasnodarsky Krai, Russia) was added. The plates were swayed on a shaker (60 rpm) to dissolve the formazan. The absorbance was measured using a microplate reader (VICTOR X5 Light Plate Reader, PerkinElmer, Waltham, MA, USA) at a wavelength of 565 nm (in order to measure formazan concentration). The results were used to construct a dose-response graph and to estimate the CC_50_ value.

### 3.9. Cancer Cell Proliferation Assays

#### 3.9.1. Cell Lines and Tumoursphere Formation

The MDA-MB-231 (RRID: CVCL_0062) and BT-549 (RRID: CVCL_1092) cell lines were purchased from ATCC and cultured in DMEM/F12 (Gibco, Life Technologies, 31330-038) supplemented with 10% FBS, 1% Pen-Strep, 1% Sodium Pyruvate, and 1% L-glutamine. Chemoresistant cell lines were established with continuous treatment for 6 months with the anticancer therapeutic agent, cyclophosphamide (Rcyclo), as previously described [67]. These cell lines were authenticated using short tandem repeat (STR) profiling within the last three years. To obtain CSC-like cells, a single cell suspension was prepared using enzymatic disaggregation (1× Trypsin-EDTA, Gibco, 25300062), and the cells were plated at a density of 10.000–12.000 cells per ml in a Cancer Stem Cell medium (C-28070, PromoCell, Heidelberg, Germany) in poly-2-hydroxyethyl methacrylate (Poly-HEMA, Santa Cruz Biotechnology, Dallas, TX, USA, sc-253284)-coated plates. Cells of the first generation (G1) were collected 10 days after seeding. The cells were transfected with corresponding compounds or DMSO (control), followed by the abovementioned procedure of tumoursphere formation. The relative numbers of tumourspheres per well were counted manually. The experiments were performed independently at least 2 times, with several replicates.

#### 3.9.2. Cancer Cell Proliferation

The survival assay was performed essentially as described earlier [77]. In short, cells were seeded into a 96-well plate, followed by treatment with increasing concentrations of the corresponding CAM-TPP derivatives for 3 days. 3-(4,5-dimethylthiazol-2-yl)-2,5-diphenyltetrazolium bromide (0.5 mg/mL MTT; Sigma-Aldrich) dissolved in media was added to each well. Following incubation for 2 h, the supernatant was carefully removed from the wells, and 100 µL of a DMSO:ethanol mix (1:1) was added to each well, followed by shaking for 10 min. The absorbance was measured at 570 nm in Bio Spec 1601, Shimadzu spectrometer. The OD570 of the DMSO solution in each well was considered to be proportional to the number of cells. The OD570 of the control (treatment without supplement) was considered to be 100%. The results were expressed as the means ± SEM. The data were analyzed using the GraphPad Prism 7 PRIZM computer software under the license of the Statistical Department (Vall’ d Hebron Hospital, Barcelona, Spain).

### 3.10. Molecular Dynamics Simulations

The structure of the *E. coli* AP–ribosome modeled in [78] was used. A cubic fragment was extracted from this structure in the same manner as in the cited work. CAM-Cn-TPPs, shown in Appendix A (atom numbering is based on [79]), were docked into this ribosome fragment using the rDock [80] package, with 1000 runs of the optimization process. Then, the pose of the sequent CAM-Cn-TPP, which shows the highest predicted affinity to the ribosome among the other poses of the same compound, was placed in the above-described fragment of ribosome. The constructed system was centered in a tetragonal cell, with dimension of 9.1 × 9.1 × 10 nm, so, when it was filled with water, the edges of the ribosome fragment were covered by a solvent layer that was at least 0.9 nm thick. During the molecular dynamics simulations, residues with at least one atom located within 0.1 nm from the edge of the simulated ribosome fragment were positionally restrained. Such an approach preserves the local conformational movability of rRNA residues, which is adequate for fitting their conformations to the binding ligand. 

The equilibrium molecular dynamics simulation of a 200 ns length was performed for every constructed system, and the coordinates of the simulated system were recorded every 25 ps, with an integration time step of 2 fs. The lengths of the covalent bonds with hydrogen atoms were limited by the LINCS algorithm [81]. The velocity rescaling thermostat, with an additional stochastic term [82] at a constant temperature of 300 K and 0.1 ps coupling time, was applied during the simulation, and the Berendsen barostat [83], with a 5 ps coupling time, was used to support the isotropic constant pressure with periodic boundary conditions. The particle mesh Ewald algorithm, with a 0.125 nm grid step and the fourth order interpolation, was used to treat long-range electrostatic interactions [84]. TIP4P*EW* water was used as a solvent. Potassium ions with optimized parameters [85] were added to neutralize the residual negative charge of the system, and they were placed near negatively charged groups [86]. To prevent the elution of magnesium and potassium counterions into the aqueous phase, part of the water molecules were randomly replaced with magnesium, potassium, and chlorine ions, setting the concentrations of MgCl_2_ to 7 mM and KCl to 100 mM. 

Canonical and modified amino acid and nucleotide residues were modeled with parm99sb [87], while CAM-Cn-TPPs were modeled with the GAFF force field [88]. Optimized three-dimensional structures and molecular electrostatic potentials of the newly parameterized residues and compounds were prepared by quantum chemical Hartree-Fock calculations using the 6–31G* basis set. Partial charges were evaluated with the RESP model [89]. 

GROMACS [90,91] software version 5.1.4 was used to simulate the molecular dynamics simulations and analyze the obtained trajectories, including an analysis of the hydrogen bonds and stacking interactions, which were performed in the same way as in [92], clustering of frames using the GROMOS [93] method, and calculation of the energy of noncovalent interactions: $E_{noncov}=E_{VdW}+E_{Coulomb}$.

## 4. Conclusions

Based on our recent findings from the study of CAM-C4-TPP [51], in this work, we set out to create dual-acting antimicrobial compounds, the structure of which would include an amphenicol fragment of CHL and a TPP cation, connected by linkers of different lengths (CAM-Cn-TPP). We synthesized CAM-C10-TPP and CAM-C14-TPP and examined their ribosomal binding and translational inhibitory properties, as well as their effects on the bacterial membrane. CAM-C10-TPP and CAM-C14-TPP bind to the bacterial ribosome, and their affinity depends on the length of the linker connecting the two chromophores. New CHL analogs inhibit protein synthesis in vitro, such as CAM-C4-TPP, allowing for the formation of multiple peptide bonds, but the mode of action of CAM-Cn-TPP in the bacterial ribosome differs from the site-specific action of CHL, as previously observed for CAM-C4-TPP. Moreover, we showed that CAM-C10-TPP more strongly inhibited the growth of the Gram-positive bacteria, *Staphylococcus aureus*, *Listeria monocytogenes*, *Bacillus subtilis*, and *Micobacterium smegmatis*, than CHL, and both CAM-C10-TPP and CAM-C14-TPP inhibited some CHL-resistant bacterial strains. At the same time, we found that CAM-C10-TPP and CAM-C14-TPP caused a significant decrease in the membrane potential in *Bacillus subtilis* cells, and, apparently, this effect makes the main contribution to the antibacterial action of the new compounds. Thus, we have shown that based on a ribosome antibiotic (CHL) and a penetrating cation (TPP), it is possible to obtain antimicrobial agents that act simultaneously on the bacterial ribosome and on the bacterial membrane. Such an approach can be developed in the future to create new antibacterial agents by reducing the toxicity of the compounds, for example, using other more physiologically penetrating cations and possibly new antiproliferative agents as well.

## Figures and Tables

**Figure 1 antibiotics-10-00489-f001:**
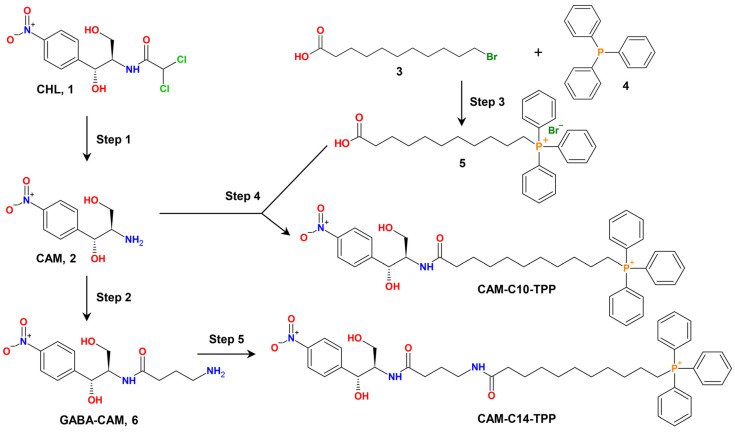
Scheme of the chemical synthesis of triphenylphosphonium (TPP) analogs of CHL: CAM-C10-TPP and CAM-C14-TPP. Step 1: 1M hydrochloric acid (HCl) at 100 °C for 2 h. Step 2: (1) Boc-GABA-OSu, dimethylformamide (DMF), and diisopropylethylamine (DIPEA) at 25 °C for 24 h; and (2) trifluoroacetic acid (TFA) at 25 °C for 30 min. Step 3: benzene at 85 °C for 12 h. Step 4: (1) **5**, 1-hydroxysuccinimide (HOSu), *N,N*′-dicyclohexylcarbodiimide (DCC), and dichloromethane (CH_2_Cl_2_) at 0 °C for 2 h, then overnight at RT; (2) **2**, DIPEA, DMF, and stirring at RT for 5 h, then overnight at 4 °C. Step 5: (1) **5**, HOSu, DCC, and dichloromethane (CH_2_Cl_2_) at 0 °C for 2 h, then overnight at RT; (2) **6**, DIPEA, DMF, and stirring at RT for 5 h.

**Figure 2 antibiotics-10-00489-f002:**
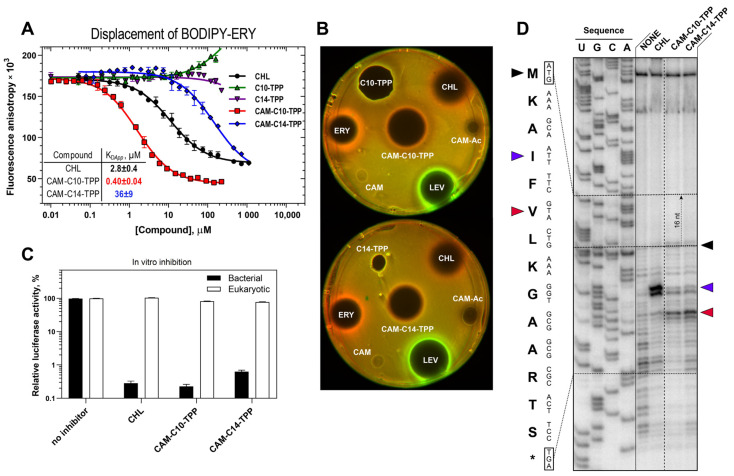
Binding affinity to bacterial ribosomes and the inhibition of protein synthesis by CAM-C10-TPP and CAM-C14-TPP. (**A**) A competition-binding assay to test the displacement of fluorescently labeled analogs of the erythromycin, BODIPY-ERY, from *E. coli* 70S ribosomes in the presence of increasing concentrations of CHL (black circles), CAM-C10-TPP (red squares), CAM-C14-TPP (blue rhombus), decyl(triphenyl)phosphonium bromide (C10-TPP, green triangles), or tetradecyl(triphenyl)phosphonium bromide (C14-TPP, purple triangles), measured by fluorescence anisotropy. All reactions were repeated four times. Error bars represent the standard deviation. The resulting values for the apparent dissociation constants (K_Dapp_) are shown on the plot. (**B**) Testing of the CAM-C10-TPP and CAM-C14-TPP activity using *E. coli BW25113 ΔtolC pDualrep2* reporter strain. The induction of the red fluorescent protein expression (green halo around the inhibition zone, pseudocolor) is triggered by DNA-damage, while the induction of Katushka2S protein (red halo) occurs in response to ribosome stalling. Levofloxacin (LEV), erythromycin (ERY), chloramphenicol (CHL), N-acetyl-chloramphenicol amine (CAM-Ac), C10-TPP, and C14-TPP are used as the controls. (**C**) The inhibition of protein synthesis by 30 µM of CHL, CAM-C10-TPP, or CAM-C14-TPP in vitro in the cell-free bacterial (black columns) and eukaryotic (transparent columns) transcription–translation coupled system. The relative enzymatic activity of in vitro synthesized firefly luciferase is shown. The error-bars represent the standard deviations of the mean of three independent measurements. (**D**) Ribosome stalling by CAM-Cn-TPP on *trpL* mRNA in comparison with CHL, as detected by a reverse-transcription primer-extension inhibition (toeprinting) assay in a cell-free translation system. The nucleotide sequences of *trpL* mRNA and their corresponding amino acid sequences are shown on the left. The black arrowhead marks the translation arrest at the start codon, while the colored arrowheads point to drug-induced arrest sites within the coding sequences of the mRNAs used. Note that due to the large size of the ribosome, the reverse transcriptase used in the toeprinting assay stops 16 nucleotides downstream of the codon located in the P-site.

**Figure 3 antibiotics-10-00489-f003:**
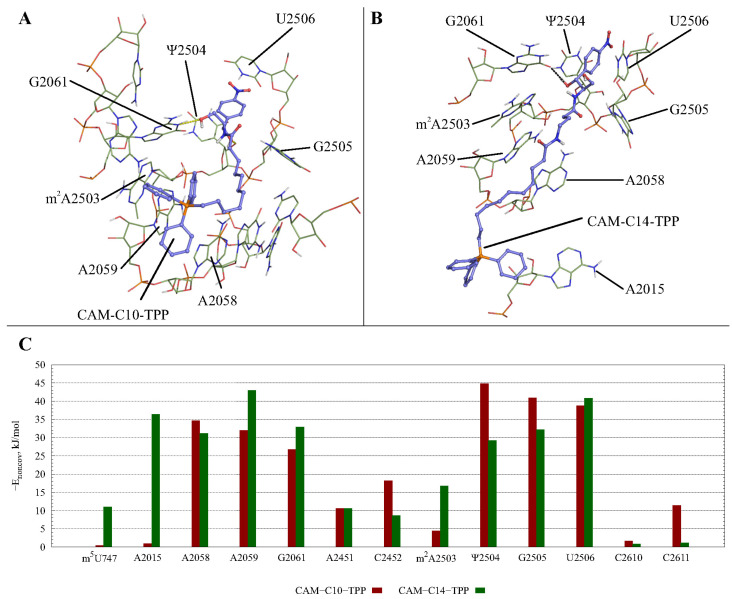
Interactions between CAM-C10-TPP (**A**), CAM-C14-TPP (**B**), and *E. coli* A,A/P,P–ribosome, obtained using MD simulations. Hydrogen bonds are shown by black dashes. CAM-Cn-TPP nitrophenyl fragment is immersed in the “hydrophobic cavity” formed by the Ψ2504 and U2506 bases, forming stacking interactions with them. The arrangement of hydrogen bonds for the CAM-C10-TPP complex (**A**) corresponds to the non-canonically linked CHL [59]. For CAM-C14-TPP (**B**), a stable hydrogen bond between the O1-hydroxyl group of the CAM residue and O6 of G2061 is observed. The TPP fragment of CAM-C10-TPP interacts with the “hydrophobic cavity” of the macrolide binding site, forming developed hydrophobic contacts with the A2058, A2059, and C2610 bases. The long C14-linker in CAM-C14-TPP appears in the “hydrophobic cavity” between the A2058 and A2059 bases, and the TPP fragment is located deeper in the NPET adjacent to the residue, A2015. (**C**) The energy of the noncovalent interactions between CAM-C10-TPP (red columns) or CAM-C14-TPP (green columns) and the neighboring 23S rRNA residues of the *E. coli* ribosome, which are in the canonical A,A/P,P–state. *E_noncov_* is shown with a negative sign for improved readability.

**Figure 4 antibiotics-10-00489-f004:**
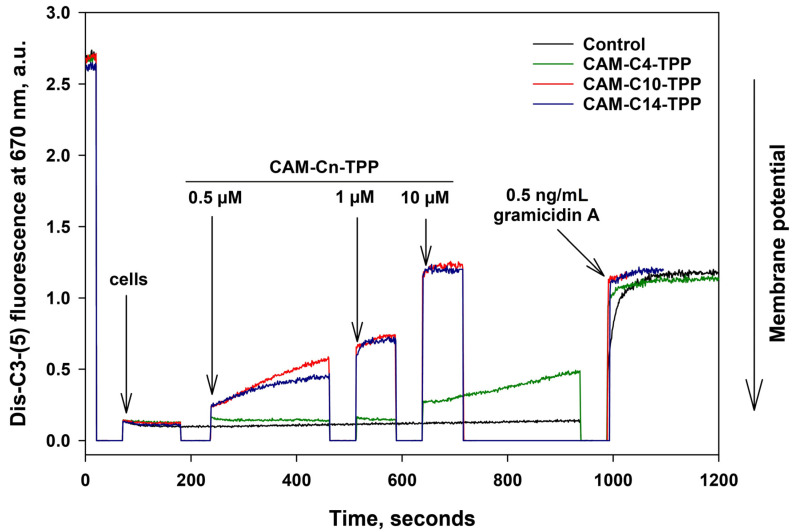
Dose-dependent effect of CAM-Cn-TPP on the kinetics of the membrane potential of *B. subtilis* cells, as assessed by DiS-C3-(5) (10 µM) fluorescence in a PBS buffer. To reach the desired concentrations, appropriate amounts of CAM-Cn-TPP were added at different moments, which are marked by the arrows. The Gramicidin A concentration was 0.5 ng/mL.

**Figure 5 antibiotics-10-00489-f005:**
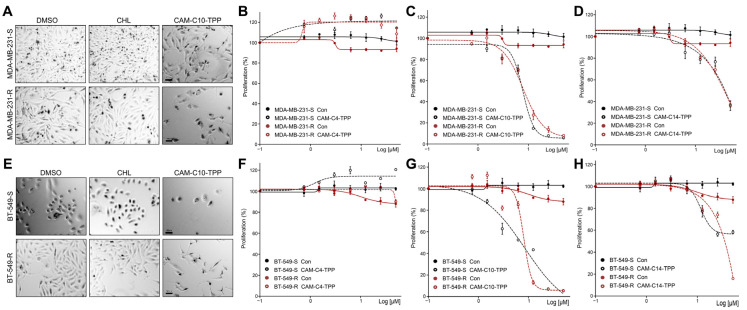
Effect of CAM-TPP derivatives on the cell viability of TNBC cells. MDA-MB-231 (**A**–**D**) and BT-549 (**E**–**H**) TNBC-sensitive (S) and chemoresistant (R) cells were grown at a 60% confluence for 3 days with the indicated concentrations of CAM-TPP derivatives. Representative images (**A**,**E**) were obtained at 40× magnification. The scale bar is 10 µm. Cells were subjected to viability assays. The results represent the mean of 3 independent experiments. The data indicate the mean ± SEM. The *p*-values, all relative to controls, were statistically significant (*p* < 0.05).

**Figure 6 antibiotics-10-00489-f006:**
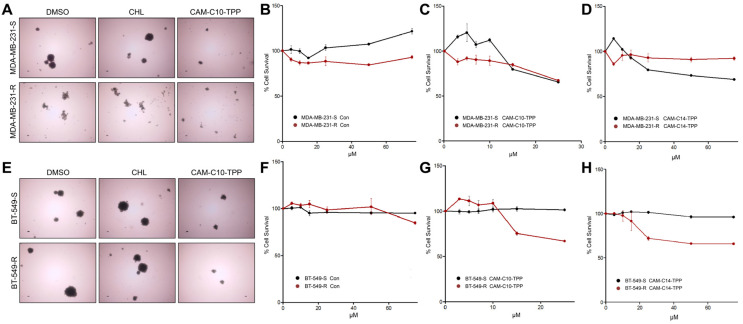
Effect of CAM-TPP derivatives on the cell viability of CSC-like TNBC cells. MDA-MB-231 (**A**–**D**) and BT-549 (**E**–**H**) TNBC-sensitive (S) and chemoresistant (R) cells were grown under non-adherent conditions for 5 days in the presence of CAM-TPP derivatives to form spheroids. Representative images (**A**,**E**) of CAM-C10-TPP-treated cells show a decrease in spheroid size. CSC-like cells were then tested for viability as before. The results are the mean of 3 independent experiments. The data indicate the mean ± SEM. The *p*-values, all relative to controls, were statistically significant (*p* < 0.05).

**Table 1 antibiotics-10-00489-t001:** Suppression of the growth of Gram-positive bacteria by CAM-Cn-TPPs. Values of the minimal inhibitory concentration (MIC, µM) are shown ^1^.

	*Staphylococcus aureus*	*Listeria* *monocytogenes*	*Bacillus* *subtilis*	*Mycobacterium smegmatis*
CHL	60	25	12	6
CAM-C10-TPP	5	6	2	2
CAM-C14-TPP	12	17	12	8
C10-TPP	2	5	2	<2
C14-TPP	1.6	2	8	4

^1^ MIC values were determined using the double-dilution method. The MIC for each compound was determined in triplicate in two independent sets.

**Table 2 antibiotics-10-00489-t002:** Suppression of the growth of *E. coli* strains, with the deletion of the *tolC* gene and the harboring CHL acetyltransferase (*cat*) gene (*E.coli* ∆*tolC-CAT*) or *B. subtilis* CHL-resistant strains (*B. subtilis pHT01-cat*) and the harboring methyltransferase Cfr *(cfr)* gene (*B. subtilis pHT01-cfr*) by CAM-Cn-TPPs. The values of a minimal inhibitory concentration (MIC, µM) are shown ^1^.

	*E. coli* ∆*tolC*	*E. coli* ∆*tolC-CAT*	*Bacillus* *subtilis*	*B. subtilis pHT01-cat*	*B. subtilis pHT01-cfr*
CHL	2.8	>360	12	180	90
CAM-C10-TPP	3	25	2	6	6
CAM-C14-TPP	12.5	12.5	12	12.5	6
C10-TPP	3	3	2	1.6	0.8
C14-TPP	1.6	6	8	1.6	0.8

^1^ The MIC values were determined using the double-dilution method. The MIC for each compound was determined in triplicate in two independent sets.

**Table 3 antibiotics-10-00489-t003:** Growth inhibition by CAM-Cn-TPPs in relation to a number of cell lines according to the MTT assay. Values of a 50% growth inhibition concentration (GI50, µM) are shown.

	*HEK293T*	*MCF7*	*A549*	*VA13*
CHL	>50	>50	>50	>50
CAM-C10-TPP	0.62 ± 0.04	1.0 ± 0.1	0.7 ± 0.1	2.8 ± 0.5
CAM-C14-TPP	3.6 ± 0.5	5.8 ± 0.8	2.9 ± 0.4	5.2 ± 0.9
C10-TPP	0.08 ± 0.01	0.21 ± 0.03	0.07 ± 0.01	0.27 ± 0.05
C14-TPP	0.03 ± 0.02	0.02 ± 0.01	0.025 ± 0.009	0.07 ± 0.05
Doxorubicin	0.007 ± 0.001	0.04 ± 0.01	0.04 ± 0.01	0.18 ± 0.04

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
