# Peer review of "Triphenilphosphonium Analogs of Chloramphenicol as Dual-Acting Antimicrobial and Antiproliferating Agents"

_antibiotics, 2021, doi:10.3390/antibiotics10050489_

Round 1

Reviewer 1 Report

Overall, the paper is well-written. There are a few grammar/spelling issues, so a final check of language would be recommended. There are also a few corrections that I recommend:

  • Please define DLCs (line 463)
  • There are quite a few large gaps in the paper (i.e., pages 13, 15) – figures need to fit into the paper better.
  • The synthesized compounds are interesting but much too toxic I believe. The conclusions support this, but the toxicity is underplayed in Section 2.6. I do not believe you should say they can be considered as antimicrobial compounds, at least not without significant modifications. Yes they are less toxic than alkyl TPPs, but not by much.

Author Response

Comments and Suggestions for Authors

Overall, the paper is well-written. There are a few grammar/spelling issues, so a final check of language would be recommended.

Response:       We are grateful to the Reviewer for the careful reading of the manuscript and valuable comments. We have carefully checked the grammar and spelling of the English text. The text of the manuscript was changed and the corresponding corrections were made in the word version in the tracking mode.

There are also a few corrections that I recommend:

  • Please define DLCs (line 463) –

Response:       Corrected, the abbreviation is deciphered: “delocalized lipophilic cation containing compounds” (line 469 in new pdf file)

  • There are quite a few large gaps in the paper (i.e., pages 13, 15) – figures need to fit into the paper better.

Response:       We agree with the comment and in the updated text we present Figures 5 and 6 in a more readable format (lines 490 and 502)

  • The synthesized compounds are interesting but much too toxic I believe. The conclusions support this, but the toxicity is underplayed in Section 2.6. I do not believe you should say they can be considered as antimicrobial compounds, at least not without significant modifications. Yes they are less toxic than alkyl TPPs, but not by much.

Response:       We completely agree with the Reviewer and we have removed the phrase “The lower toxicity of the synthesized analogs against mammalian cells, compared to alkyl-TPPs, is a favorable property in terms of considering CAM-Cn-TPPs as antimicrobial compounds” from the text (Section 2.6, before the table 3, lines 454-456 in the previous version of the manuscript).

Reviewer 2 Report

The manuscript described anti-bacteria agents. The authors developed triphenilphosphonium analogs of chloramphenicol showing antimicrobial and antiproliferating effects. In addition, these agents reduced breast cancer cell proliferation based on mitochondria. Thus, these findings will be useful not only for bacteria but also cancers.  Therefore, the manuscript is not too excellent to be published. In other words, the manuscript is so excellent that it should be published.

Comments

(1) Are there any pharmacokinetic differences in vivo between CHL and CAM-Cn-TPP?

(2) Do TPP derivatives affect mitochondria in normal cells?

(3) How are CAM-Cn-TPP derivatives delivered selectively into cancer cells?

(4) In Figure 1, 2 of “CH2Cl2” should be subscript. “hours” and “h” should be used as the same style. “N,N'-” should be italic font.

(5) In line of 201, “We’ve” should be “We have”.

(6) In line of 545, 3 of “CHCl3” should be subscript.

That is all.

Author Response

Comments

The manuscript described anti-bacteria agents. The authors developed triphenilphosphonium analogs of chloramphenicol showing antimicrobial and antiproliferating effects. In addition, these agents reduced breast cancer cell proliferation based on mitochondria. Thus, these findings will be useful not only for bacteria but also cancers.  Therefore, the manuscript is not too excellent to be published. In other words, the manuscript is so excellent that it should be published.

(1) Are there any pharmacokinetic differences in vivo between CHL and CAM-Cn-TPP?

Response:       We are thankful to the reviewer for such a careful and critical reading of the manuscript and valuable questions.

            So far, we have not investigated the pharmacokinetics of CAM-Cn-TPP compounds, although this is certainly of interest in terms of creating therapeutic agents. Most likely, the pharmacokinetics of the described compounds will differ from that of chloramphenicol. And, most likely, judging by our research, it will be determined mainly by the alkylTPP fragment. Among the many compounds of alkylTPP, the pharmacokinetics of conjugates with antioxidants (e.g, SkQ and MitoQ) have been most fully studied. According to various sources, these compounds, depending on the method of administration and organisms, can accumulate in heart and tumor tissues, liver, kidney and gut with almost complete clearence within 24-48 h (Zielonka, 2017 (doi:10.1021/acs.chemrev.7b00042); Wang, 2020 (DOI: 10.1002/cmdc.201900695). Chloramphenicol has been found in addition to the mentioned organs in significant amounts in the cerebrospinal fluid. It can be assumed that the metabolites of CAM-Cn-TPPs will also differ from the metabolites of chloramphenicol. It is impossible to exclude the appearance of metabolites associated with the reduction of the nitro group or the acylation of hydroxyl groups, as in the case of chloramphenicol (Bories, 1994 (https://doi.org/10.3109/03602539408998326). It is likely that the amide bond will break down during the metabolism to form chloramphenicol amine (which is one of the metabolites of chloramphenicol). At the same time, toxic metabolites associated with reactions and transformations of the dichloroacetyl residue should not be formed, and, possibly, as a result, there would be no side effects associated with the inhibition of cytochrome P450 (Correia, 2005 in: Cytochrome P450: Structure, Mechanism, and Biochemistry, 3e, edited by Paul R. Ortiz de Montellano, Kluwer Academic / Plenum Publishers, New York, 2005 (Chapter 7) and the appearance of aplastic anemia.

(2) Do TPP derivatives affect mitochondria in normal cells?

Response:       Since TPP directs any conjugated compound to the mitochondria, it is assumed that TPP derivatives affect both mitochondrial functions not only in cancer cells but also in normal cells. We are in the process of selecting derivatives that discriminate their effects between healthy and cancer cells. In the corrected version we propose a supplementary Figure (Figure S3) where in particular the effects of CAM-C10-TPP on non-cancer (normal fibroblasts) and cancer TNBC cells are compared. At low concentrations the growth of cancer cells is inhibited better than that of healthy cells. Also note that this effect disappears at high concentrations. In the Results and Discussion section (2.7), we added a paragraph:

“We also compared the effects of the obtained compounds on the suppression of normal cell growth. Preliminary results with CAM-C10-TPP showed that normal human fibroblasts were less affected by submicromolar concentrations of TPP-CAM derivative than cancer cells (Figure S3). This suggests a pharmacological window to discriminate between healthy and cancer cells.” - lines 508-512

(3) How are CAM-Cn-TPP derivatives delivered selectively into cancer cells?

Response:       CAM-Cn-TPP derivatives cannot be selectively delivered to cancer cells.  However, there are a number of factors that contribute to a stronger effect of the derivatives on cancer cells specifically:

(i) cancer cells divide much faster than non-cancerous cells and in low concentrations CAM-Cn-TPP derivatives lead to mitochondrial dysfunction and thus death of cancer cells.

(ii) the metabolic profiles of normal, cancerous, resistant cancer cells and cancer stem cells differ (see Lleonart, 2018 (PMID: 28683561, DOI: 10.1089/ars.2017.7223); Abad, 2018 P(MID: 30373788, DOI: 10.1074/mcp.RA118.001102); Abad, 2020 (PMID: 32592934, DOI: 10.1016/j.bbadis.2020.165886) and one would expect less effect of CAM-Cn-TPP derivatives on healthy cells with well functioning mitochondria.

(4) In Figure 1, 2 of “CH2Cl2” should be subscript. “hours” and “h” should be used as the same style. “N,N'-” should be italic font.

Response:       We thank the Reviewer for the typos and inaccuracies noticed. Typos and inaccuracies are corrected (lines 199-203)

(5) In line of 201, “We’ve” should be “We have”.

Response:       After checking the correctness of the English spelling and grammar, the tense of the verbs in this phrase is changed: “We chose…” (line 207)

(6) In line of 545, 3 of “CHCl3” should be subscript.

Response:       Corrected (line 556-557)